# Temporal Monte Carlo Dropout for Robust Uncertainty Quantification: Application to Point-of-Care Ultrasound-guided Nerve Blocks

**Nishanth Thumbavanam Arun**[1]                    NTHUMBAV@ANDREW.CMU.EDU

**Leonard Weiss**[2]                                          WEISSLS2@UPMC.EDU

**Andrew Schoenling**[2]                                  SCHOENLINGAJ@UPMC.EDU

**Marek Radomski**[2]                                     RADOMSKIMA@UPMC.EDU

**Frank Guyette**[2]                                           GUYEFX@UPMC.EDU

**Napoleon Roux**[3]                              NAPOLEON.P.ROUX.MIL@HEALTH.MIL

**Brittany Daley**[3]                           BRITTANY.S.DALEY.CTR@HEALTH.MIL

**Michael J Morris**[3]                     MICHAEL.J.MORRIS34.CIV@HEALTH.MIL

**Howie Choset**[1]                                              CHOSET@CMU.EDU

**John Galeotti**[1]                                          JGALEOTTI@CMU.EDU

[1] *Carnegie Mellon University, Pittsburgh, PA*

[2] *University of Pittsburgh Medical Center, Pittsburgh, PA*

[3] *Brooke Army Medical Center, Fort Sam Houston, TX*

## Abstract

Accurate needle placement during nerve block procedures is essential for safe and effective anesthesia and pain management. However, tracking needles and nerves in an austere setting using Point of Care Ultrasound (POCUS) can be challenging due to the complexity of the surrounding anatomy, the lack of real-time feedback and limited image quality. In this paper, we propose a method for segmenting these structures and estimating the pixelwise uncertainty using a novel approach: Temporal Monte Carlo Dropout. We demonstrate the effectiveness of our approach in POCUS with a stable probe, where it provides robust uncertainty estimates in challenging imaging scenarios while simultaneously tracking the needle accurately. Our method obtains an 84% similarity score with uncertainty estimates obtained from Monte Carlo Dropout with an 8x decrease in computational complexity without compromising segmentation performance. Importantly, it can be easily integrated into existing POCUS workflows on portable devices and has the potential to benefit medical practitioners and patients alike.

**Keywords:** POCUS AI, Bayesian Inference, Anatomic Segmentation, Needle, Nerve Block

## 1. Introduction

Point-of-care ultrasound (POCUS) has emerged as a powerful tool for rapid, accurate, and portable diagnosis and treatment, particularly in scenarios where traditional imaging modalities are not available or are impractical to use. POCUS can improve patient outcomes, reduce costs, and increase efficiency in the healthcare system(Kuo et al., 2020; Magalhães et al., 2020; Smallwood and Dachsel, 2018). The provision for real-time visualization makes it an ideal tool for guiding nerve block procedures. Historically, nerve block needle placement has been guided by methods that are invasive, imprecise, and costly due to the need for specialized equipment and personnel(Hadzic et al., 2003) (Choquet et al., 2012)

Ensuring safety and efficacy of deep learning models in such ultrasound-guided procedures demands precise uncertainty estimation. Traditional Bayesian uncertainty estimation

methods, such as Markov Chain Monte Carlo sampling(Van Ravenzwaaij et al., 2018), ensemble methods(Vrugt and Robinson, 2007), and Monte Carlo dropout(Camarasa et al., 2020), are computationally expensive, limiting real-time feedback. Our Temporal Monte Carlo Dropout method overcomes this by sampling once per frame with varying dropout configurations across frames, maintaining reliable uncertainty estimates while reducing complexity and not compromising on segmentation performance.

## 2. Methods

The work was performed under IRB approvals from all investigators' home institutions. The Peripheral Nerve Block dataset, comprising of recorded Adductor Canal (AC) block procedures, was collected at the Brooke Army Medical Center (BAMC) using a butterfly ultrasound probe and anonymized for our access and use. We examined 15 different patient recordings where needle placement was within 0.5 cm of the nerve and 16 different patient recordings where needle placement was beyond 1 cm from the nerve. We include six additional anonymized femoral nerve block clips (labelled Negative) from the University of Pittsburgh Medical Center (UPMC) Department of Emergency Medicine database.

We employ a Bayesian 3D U-Net encoder-decoder architecture for segmentation, using temporal volumes as inputs(Kendall and Gal, 2017). The model produces two outputs, the predictive mean $\hat{\mu}$ and variance $\hat{\sigma}^2$. We train the model with stochastic cross entropy loss accounting for aleatoric uncertainty(Kendall and Gal, 2017). The predicted output for a frame $x_i$ is given by $[\hat{\mu}, \hat{\sigma}^2_{aleatoric}] = f_\theta(x)$, where $f_\theta$ denotes the model parametrized by the weights $\theta$. To obtain the epistemic uncertainty maps, we use our novel temporal Monte Carlo dropout, which involves performing inference with each frame separately. The uncertainty estimate for each pixel in a frame is obtained as the variance of $\hat{\mu}$ across M frames, each run with different dropout configurations $\sigma^2(x_i) = \frac{1}{M}\sum_{i=1}^{M}(\hat{\mu_i} - \bar{\mu})^2$. The model was trained with the Adam optimizer(Kingma and Ba, 2014) with a learning rate of 1e-4 and early stopping(Yao et al., 2007)

## 3. Results and Discussion

We compare the uncertainty maps generated using temporal MC dropout across M frames with nontemporal MC dropout across N samples per frame where nontemporal MC dropout refers to the standard way of computing MC dropout with $\sigma^2(x_i) = \frac{1}{N}\sum_{i=1}^{N}(\hat{\mu_i} - \bar{\mu})^2$ for N MC samples per frame

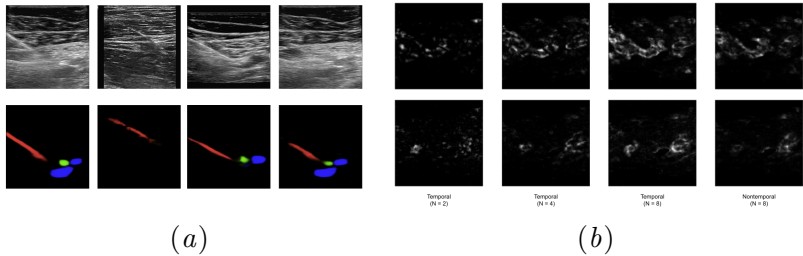

$(a)$ $(b)$

**Figure 1:** a) Example input frames (top) with their corresponding segmentations (bottom : Red - Needle, Green - Nerve, Blue - Vessel) and b) Needle (top) and nerve (bottom) uncertainty maps with different MC configurations

We use Structural SIMilarity (SSIM)(Hore and Ziou, 2010) to compare the uncertainty maps, as SSIM has been shown to capture structural information accurately and corre-

late well with human perception(Abdar et al., 2021). We show a few visualizations of the segmentation and uncertainty maps in different configurations in Fig 1 and the SSIM comparisons in Fig 2. Our results suggest a strong correlation between Nontemporal Monte Carlo (MC) sampling with N = 8 and temporal MC sampling with M = 8. This finding indicates that our approach is effective in generating structurally similar uncertainty maps while reducing the computational burden of running the model multiple times on a given image. We validate that sampling once per frame maintains segmentation performance by comparing our method with popular UNet variants using needle tip error and needle/nerve detection as segmentation metrics, in table 1. Our method outperforms most alternatives and nearly matches the nontemporal variant with 8 MC samples per frame while reducing computational burden by a factor of $1/8$.

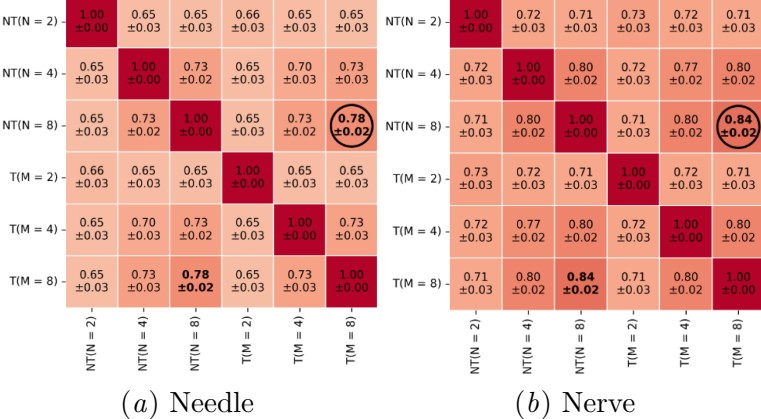

$(a)$ Needle $\qquad\qquad$ $(b)$ Nerve

**Figure 2:** Mean +/- Std SSIM scores over the test set comparing needle uncertainty maps generated with different values of N (nontemporal monte carlo dropout samples) and M (temporal monte carlo samples across frames) (NT - Nontemporal T - Temporal)

**Table 1:** Segmentation performance

| Model | Tip Error(cm) | Needle Detection | Nerve Detection |
|---|---|---|---|
| **Bayesian 3D UNet (ours)** | 0.23±0.18 | **100%** | **66%** |
| Bayesian 3D UNet (nontemporal) | **0.22±0.17** | **100%** | **66%** |
| 3D UNet | 0.29±0.18 | 99% | 45% |
| Bayesian 2D UNet | 0.32±0.26 | 88% | 61% |
| 2D UNet | 0.31±0.21 | 97% | 49% |
| UNet+LSTM | 0.32±0.18 | 87% | 50% |

## 4. Conclusion

We presented a novel method to extract uncertainty maps with significant computing advantages over alternatives, and showed that we did not compromise on output quality and model accuracy. Further work can look at extending this algorithm to settings with moving probes and to other imaging modalities.

## Acknowledgments

This material is based upon work supported by the Defense Advanced Research Projects Agency (DARPA) under Agreement No. HR00112190075

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
