# OpenReview forum: "Temporal Monte Carlo Dropout for Robust Uncertainty Quantification: Application to Point-of-Care Ultrasound-guided Nerve Blocks"
_MIDL.io/2023/Short_Paper_Track — MIDL 2023 Short paper track Poster_

### Official Review · Reviewer_r6sZ · 2023-04-19
**Strong paper on needle/analomy tracking of ultrasound imagery**

**Rating:** 8
**Confidence:** 4

**Review:**

This paper presents a technique to segment needles and anatomical structures Ultrasound-guided nerve blocks anesthesia. A novel approach is proposed called "Temporal Monte Carlo Dropout" that provides robust uncertainty estimates in challenging imaging scenarios while maintaining accuracy and reducing computational complexity. Validation was performed on N=15+16 patients.

Pros:
* The proposed TMCD method reduces the computational complexity by a factor of 8 without compromising segmentation performance.
* The method demonstrates the effectiveness of providing robust uncertainty estimates in challenging imaging scenarios.
* The approach can be easily integrated into existing workflows on portable devices, making it more accessible to medical practitioners.
* The paper is well-structured and provides clear and concise explanations of the methodology and results.

No real cons but rather ideas for related and future work:
* Investigate the performance of the proposed TMCD method in settings with moving probes to understand how the method can be adapted to various clinical scenarios.
* Deeper analysis on the trade-offs between segmentation performance, uncertainty estimation, computational complexity and performance.
* Evaluate the performance of the proposed method in a clinical setting to obtain real-world feedback from medical practitioners and better understand its practical implications.
* Provide implementation details in some public repository

---

### Official Review · Reviewer_vUJy · 2023-04-24
**Authors could compare to Transformer U-Net. Uncertainty maps may overwhelm clinicians with existing data.**

**Rating:** 6
**Confidence:** 4

**Review:**

This paper introduces a new approach to accurately track needles and nerves during nerve block procedures using Point of Care Ultrasound (POCUS), even in challenging imaging scenarios. The proposed method segments the structures and estimates pixelwise uncertainty using a novel approach called Temporal Monte Carlo Dropout, which provides robust uncertainty estimates while accurately tracking the needle. The method is easily integrable into existing POCUS workflows on portable devices, potentially improving the safety and effectiveness of anesthesia and pain management for both medical practitioners and patients.

However, the authors could have compared their method with more recent segmentation techniques such as the Transformer U-Net model. Additionally, while uncertainty maps may provide valuable information, clinicians may find it overwhelming as they already have access to various data during procedures.